# The Selected Topics for Comparison in Visegrad Four Countries

**Anna Kowalska [1], Jaroslav Kovarnik [2,*], Eva Hamplova [2]**  **and Pavel Prazak [2]**

[1]  Wrocław University of Economics, Komandorska Street 118/120, 53345 Wrocław, Poland; anna.kowalska@ue.wroc.pl

[2]  University of Hradec Králové, Rokitanskeho 62, 50003 Hradec Kralove, Czech Republic; eva.hamplova@uhk.cz (E.H.); pavel.prazak@uhk.cz (P.P.)

*   Correspondence: jaroslav.kovarnik@uhk.cz; Tel.: +420-493-332-363

**Abstract:** Visegrad Group is a group of four countries in Central Europe, namely the Czech Republic, Slovakia, Poland, and Hungary. These countries share not only a similar history, but also similar economic development (measured for example by Gross Domestic Product (GDP)) and geo-political ideas. Nowadays, the economic development of every country and its competitiveness on the world market is supported by the creation of innovation (knowledge-based economy), especially from an Industry 4.0 point of view. The aim of this article is to compare the Visegrad Four (V4) from different perspectives. Firstly, the comparison of GPD development is done, next the analysis of foreign trade. The article presents the results of a comparative analysis of changes in innovativeness and competitiveness of the V4 economies over a period of 5 years. The Global Innovation Index (GII) shows the level of innovation of most countries in the world. Reports publishing GII were established thanks to the cooperation of Cornwall University with INSEAD (fr. *Institut européen d'administration des affaires*) Business School and World Intellectual Property Organization. The Summary Innovation Index (SII) was used in the European Innovation Scoreboard, as well as the Global Competitiveness Report and Global Competitiveness Index (GCI). The analysis shows that all members of V4 are so called moderate innovators. The Czech Republic begins to diverge from other member states in terms of SII, GII and it has been increasing its GCI as well. Poland occupies one of the last positions in the V4 innovation ranking, where Hungary was the weakest in terms of competitiveness in 2016. However, the mutual connection between GDP and above mentioned indexes shows relatively surprising results.

**Keywords:** GDP; foreign trade; competitiveness; innovation; Visegrad Group

**JEL Classification:** F43; O52

## 1. Introduction

The economic development of every country can be analysed from different perspectives, but one of the most frequently used indicators is Gross Domestic Product. However, this indicator can give somewhat confusing data. For example, traditional oil exporters have relatively high levels of Gross Domestic Product (GDP), but other aspects of life in these countries are usually not so great. Therefore, it is also recommended to compare different aspects of economic development, such as foreign trade (see for example Vannoorenberghe 2014), and other factors such as innovativeness and competitiveness.

Innovativeness and competitiveness are frequently used terms in today's globalized world, especially in the context of the upcoming Fourth Industrial Revolution, generally called Industry 4.0. This revolution puts more pressure on companies that must adapt their approaches to managing

operational processes. For this, innovativeness and competitiveness are very important. Both terms have been analyzed by many researchers, from different points of view; see for example (Despottović et al. 2014; Gibson and Naquin 2011; Olszańska et al. 2017; Özcelik and Taymaz 2004). Despite this fact, there exist no universal definitions of these terms.

As far as innovativeness is concerned, it can be defined as an ability of the country to produce and commercialize goods and services by using new knowledge and skills. Knowledge is the most comprehensive resource when it comes to developing wealth. Knowledge is dynamic, since it is created in social interactions amongst individuals and organizations (Grzybowska and Łupicka 2016). Another definition claims that innovativeness focuses on the potential of the country to create, improve and use innovations with the purpose of generating economic value. It is quite obvious that these definitions are not same, but they are very similar and both of them emphasize the fact that innovativeness supports economic growth. Innovativeness can be measured using several different tools, where one of the relatively frequently used is the Summary Innovation Index (SII).

The phenomenon of competitiveness is even more confusing. In simple terms, it can be explained as the effort of a country to be competitive in the world market (Kowalska 2016; Tarnowska 2014). However, there is still no generally accepted definition of such competitiveness; moreover, some authors have an opinion that the concept of macro-competitiveness does not exist. Despite such debate concerning the definition of competitiveness, it is estimated that there exists more than a hundred different forms of indicators for quantifying this phenomenon. One of them is the Global Competitiveness Index (GCI).

The aim of this article is to compare the Visegrad Four countries in terms of GDP and foreign trade, but also in terms of the innovativeness and competitiveness of the economies these countries using the SII and GCI indicators and to show the changes that have taken place in this regard in 2011–2016 (Despottović et al. 2014).

## 2. Methodology

Many authors of scientific publications compare different criterion in different countries with the aim of discovering mutual consistency, to identify differences, and to infer conclusions about possible future tendencies in global economy development. For example, using the Central and Eastern European model of capitalism, Farkas (2017) compares the market economies of the Western Balkan countries to the post-socialist European Union member states. It analyses the main institutional areas of a socio-economic system, such as product markets, innovation system, financial system, labor market and industrial relations, social protection and the educational system. Fagerberg et al. (2007) outline a synthetic framework, based on Schumpeterian logic, for analysing the question why do some countries perform much better than other countries. Four different aspects of competitiveness were identified: technology, capacity, demand, and price. The contribution of the paper was particularly to highlight the three first aspects, which often tend to be ignored because of measurement problems. The empirical analysis, based on a sample of 90 countries on different levels of development from 1980–2002, demonstrated the relevance of technology, capacity, and demand competitiveness for growth and development. Price competitiveness seemed generally to be of lesser importance. Since the collapse of state socialism in the late 1980s, the Czech Republic, Hungary, Poland, and the Slovak Republic have introduced a rather successful model of capitalism when compared with other post-socialist states. The article by Nölke and Vliegenthart (2009) identifies the key elements of the Dependent Market Economy (DME) model and discusses their interplay. DMEs have comparative advantages in the assembly and production of relatively complex and durable consumer goods. These comparative advantages are based on institutional complementarities between skilled, but cheap, labor; the transfer of technological innovations within transnational enterprises; and the provision of capital via foreign direct investment. Increased openness has also resulted in a growing inflow of FDI, both as a result of large-scale privatization programs and green-field investment by Cernat (2006). Authors Blanke and Hoffmann (2008) outline the five pillars of an upcoming European social model, although they deny

the existence of a closed system of social and economic regulatory mechanisms designed to correct the operation of markets. Yet, what exists in all western European member states is a comparatively high level of welfare state protection, and, at the European level, a certain number of elements for the development of a specific welfare state framework has been established.

The authors of this article evaluate macroeconomic indicators as categories related to the economic power of each analysed country. One of the most important indicators is foreign trade with goods and services, and its influence on global economic development of the Visegrad Four (V4) countries. An inseparable part of comparative analysis is the innovative potential of the analysed countries, where this potential is evaluated with the change of the Global Innovative Index (GII), and by the change of the Global Competitiveness Index (GCI).

The analysis of GDP development has been done through comparative analysis of this indicator. Consequently, the comparative analysis of foreign trade has been done, where foreign trade with goods and with services has been analysed separately, because foreign trade is the part of GDP formula in an open economy, and it can either contribute to GDP or worsen it.

Moreover, the comparison of these macroeconomic indicators from the V4 countries has been done through point rating. Every economy can get the number of points given by the formula:

$$y = \frac{x - x_{\min}}{x_{\max} - x_{\min}} \cdot 100 \tag{1}$$

where $y$ means the number of points, $x$ presents the value of each macroeconomic indicator for every year and every country, $x_{\min}$ is minimal value of this indicator from all countries and the whole analysed period, and finally $x_{\max}$ is the maximum one. Immediately from (1) it is clear that $y \in [0, 100]$ the value for the worst result of $x$ is $y = 0$, and the value for the best result of $x$ is $y = 0$. The coefficient y is computed by one of the possible data transformation methods called nonmetric scaling; more details can be found in (James 2016).

The analysis of innovativeness of V4 economies was based on Global Innovation Index reports presented by Cornwall University, INSEAD Business School and World Intellectual Property Organization. In addition, European Innovation Scoreboard (EIS) reports designed by the European Commission and the University of Maastricht were used. The Global Innovation Index was established in 2007. It is the average of 82 factors belonging to two groups:

1. Factors describing an environment conducive to innovation, called the "contribution of innovation", i.e., institutions, human potential, infrastructure, market development and development and quality of business operations.

2. Results measuring specific achievements in the field of innovation, referred to as "innovation results", i.e., scientific results and creative processes.

The value of this indicator since 2011 is in the range of 0–100 (where 100 means the highest possible level of innovation) (Kowalska and Tarnowska 2017).

The Summary Innovation Index (SII) consists of 27 indicators, divided into ten dimensions (i.e., human resources, attractive research, innovation-friendly environment, finance and support, enterprises, innovations, connections, intellectual assets, impact on employment and economic effects). These dimensions were assigned to one of four groups (framework conditions, investments, innovation activities and impact). The SII index is in the range of 0–1 (where 1 is the highest level of innovation) (European Innovation Scoreboard (EIS) (2017)).

The competitiveness of the economies of the Visegrad Group was determined using the Global Competitiveness Indicator (GCI) presented by World Economic Forum (WEF) in the World Competitiveness reports. Currently, the GCI index includes 114 indicators grouped in 12 pillars (i.e., institutions, infrastructure, macroeconomic environment, health and basic education, labor market effectiveness, financial market development, technological readiness, market size, business advancement and innovation). The GCI methodology takes into account the differences in the economic

progress of the analysed countries. The 12 pillars have been divided between the three stages of development, i.e., driven by factors, driven by efficiency and driven by innovation. (Schwab 2010).

Some drawings and tables in this study contain, instead of the names of individual countries, the symbol for each: for the Czech Republic (CZ), for Hungary (H), for Poland (PL), for Slovakia (SK).

## 3. Results

### 3.1. Characteristics of the Visegrad Four Countries

The countries of the Visegrad Group, or Visegrad Four, namely the Czech Republic, Slovakia, Hungary, and Poland, can be found in Central Europe. These countries share a similar history, where all of them were on the east side of the Iron Curtain, which means under the influence of the Soviet Union. All countries went through transformation in the 1990s, and all countries also joined the European Union together in 2004. Nowadays, these countries share some similar opinions, for example, in terms of the migration crisis. Because of the common history, these countries established the Visegrad Group in 1991 and have been cooperating together, even before they joined the EU. After their entrance in to the EU, they have still have been cooperating, with greater or lesser success, not only in general ways, but also in the field of the EU.

### 3.2. The Analysis of GDP Development

Based on the fact that the Czech Republic has currently around 10.5 billion inhabitants, Hungary around 9.8 billion, Slovakia around 5.4 billion, and Poland almost 38 billion, it is quite obvious that the level of GDP in billions of Euro is the highest in Poland, next in the Czech Republic, in Hungary, and then Slovakia last.

However, it is better to use the level of GDP per capita for comparison. According to this, the highest level is the Czech Republic, Slovakia is second, Hungary is currently in third place, and Poland is last. With respect to this information, it is good to add one interesting fact. Even if the development in the number of inhabitants in each country has not been steady, this number grew in the Czech Republic, and Slovakia (comparison of the number of inhabitants in the years 2000 and 2016), while in Hungary and Poland it dropped (all calculations have been made by authors based on data from Eurostat 2017b).

Deep analysis of GDP development shows that in all analysed countries, GDP had a significant decrease in the year 2009 (both in absolute value and in per capita) as a result of the global economic crisis. However, the after-crisis development is different. The Czech Republic was growing between 2009–2011, it was decreasing between 2011–2014, and it has been growing again since 2014. Moreover, it managed to exceed its pre-crisis value in the year 2011. Poland was last in 2008, it has been growing between 2009 and 2015, it exceeded its pre-crisis value in the year 2011, but it exceeded Hungary in 2012. However, it has dropped in 2016, where Hungary overtook Poland again. Hungary has been growing since 2009, with one exception in 2012. It was third before the crisis and it is now third again, since 2016. Slovakia has been growing since 2009, and it has managed to exceed its pre-crisis value within one year, already in 2010.

Table 1 shows the growth rate in GDP per capita (based on GDP and Main Components, 2017), where this calculation has been made, firstly comparing the year 2000 to the year 2016, while the second growth rate describes the after-crisis development (2009–2016), and the last growth rate describes the development since 2011. This last calculation has been made because of the next analysis, where both SII and GCI only have limited available data, and the following analysis has been made since 2011. As far as GDP development is concerned, it can be calculated also on the basis of a year-to-year growth rate, but nevertheless, the authors have decided to use growth rate with some base years (2000, after-crisis development, and 2011).

**Table 1.** Growth rate in Gross Domestic Product (GDP) per capita.

| Country | 2000/2016 Growth Rate | 2009/2016 Growth Rate | 2011/2016 Growth Rate |
|---|---|---|---|
| Czech Republic (CR) | 154.91% | 16.16% | 5.67% |
| Hungary (HU) | 128.14% | 22.26% | 13.24% |
| Poland (PL) | 129.59% | 34.50% | 11.94% |
| Slovakia (SK) | 260.44% | 25.43% | 13.91% |

The relatively weak position of the Czech Republic in the overall growth rate described in Table 1 is the result of a stronger position of this country compared to other countries (in 2000, the Czech Republic had a GDP per capita almost 6500 Euro, where Slovakia had only a little bit over 4000 Euro). However, the after-crisis development is more interesting. With respect to the Czech Republic, its growth rate is the lowest. It is partly a consequence of the strongest position of the V4 countries, but partly also because of unstable development after the crisis with several drops. The highest growth rate after the crisis is Poland, but compared to 2011, Poland is in third position. It means that the decrease of Poland in the crisis year was really strong and this country managed to overcome pre-crisis year in 2011. Since then, the growth rate has not been so strong. Slovakia was in last position in GDP per capita in 2000, but it had been growing significantly, and it was in second place in the pre-crisis year. Moreover, it has been growing after the crisis as well, and is in second place, and if this development continues, we can expect that it will be equal to the Czech Republic. As far as Hungary is concerned, its relatively strong growth rate is a consequence of relatively weak values of GDP per capita.

The relationship between GDP per capita and the growth rate between 2011 and 2016 is described in Figure 1. Figure 1 demonstrates that the Czech Republic has the highest level of GDP per capita, but the lowest growth rate, where other members of the V4 have relatively high growth rates, and moreover, Slovakia has a relatively high level of GDP per capita as well.

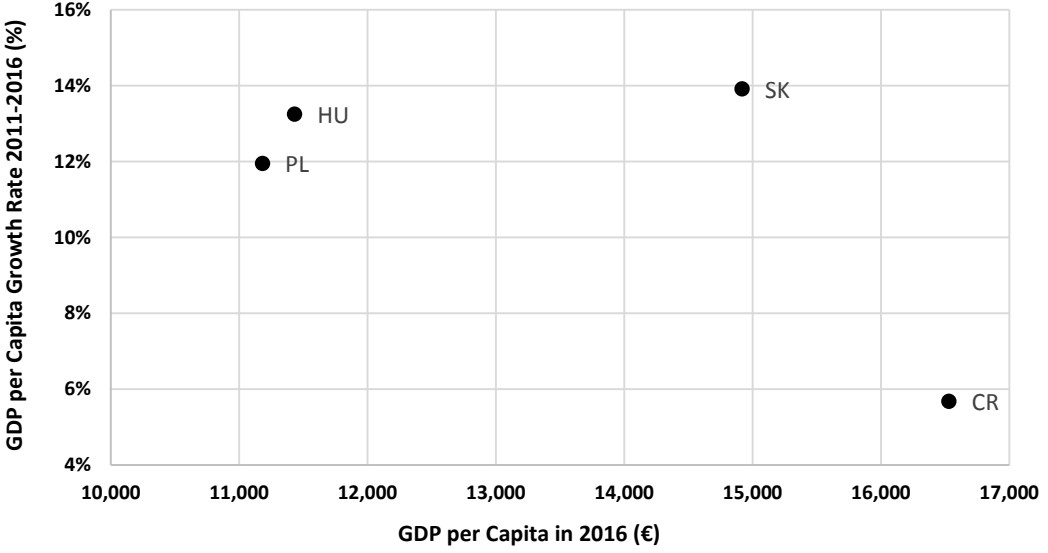

**Figure 1.** Relation between GDP per capita (2016) and growth rate (2011–2016). SK, Slovakia; HU, Hungary; PL, Poland; CR, Czech Republic.

### 3.3. Foreign Trade Development—Trade with Goods

The analysis of foreign trade development shows some different results to that of GDP development. Calculation has been done based on Eurostat 2017a, where more detailed information can be found, for example, in Kovárník and Hamplová 2016.

The first interesting fact is that all countries had a negative trade balance in the first analysed year (2000), which means they had higher imports than exports. Even if the current trade balance (in 2016,

in 2015 in Poland) is positive in all analysed countries, the development has still been quite irregular. The highest surplus in 2016 was the Czech Republic, where this surplus is just over 9000 billion. For mutual comparison, the surpluses have been recalculated again per capita.

The Czech Republic had the worst position in 2000 (the highest deficit per capita, while in absolute amount, Poland had the worst result), but it has been growing (with few exceptions) and currently the Czech Republic has the biggest surplus of all V4 countries. The exact opposite development has been seen in Slovakia. This country had the best result of the V4 countries in 2000 (both in absolute amount and per capita), but it has the worst result in 2016 in absolute amount, and the second worst in terms of net balance per capita (in comparison with Poland in 2015). A really interesting fact is that in 2009, during the economic crisis, the net balance was increasing in all analysed countries. Figure 2 describes again the relation between the values of net balance of trade with goods in 2016 and the growth rate between 2011 and 2016.

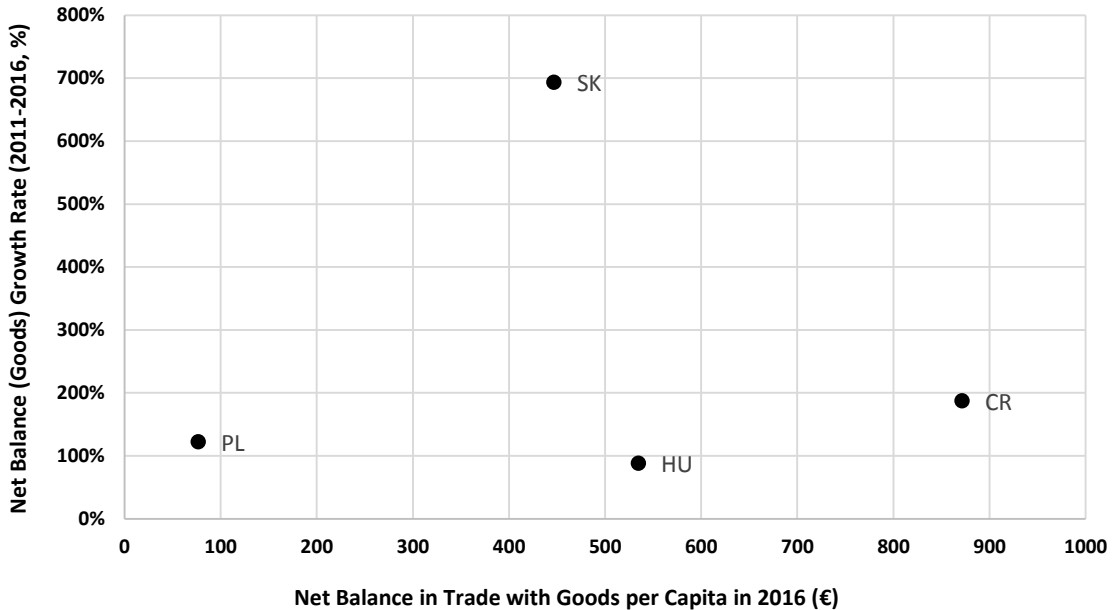

**Figure 2.** Relation between net balance in trade with goods per capita (2016) and growth rate (2011–2016).

Figure 2 describes relatively different results than that of GDP (Figure 1). The Czech Republic has the highest level of net balance per capita, but it has second highest growth rate (almost 200%). However, the second highest net balance per capita is Hungary, but this country has the lowest growth rate (less than 100%). Slovakia has the third highest net balance per capita, but it also has a significantly higher growth rate (almost 700%). On the other hand, it has to be mentioned that Slovakia was in a small deficit in 2011, and it has a relatively high surplus in 2016, therefore the growth rate between these two years is so extraordinarily high. The lowest value of net balance per capita in 2016 is Poland, where this country has also relatively low growth rates. It appears that foreign trade with goods is not as important in this country as it is in the other V4 member states.

However, the export/GDP ratio shows that Slovakia currently has the highest value of this indicator. This indicator measures the openness of a particular economy and it is relatively surprising that even if Slovakia has a relatively bad result in terms of net balance with goods, it can be considered as a really open economy, where this share is more than 84% in 2016. Another open economy is Hungary (more than 73%) and also the Czech Republic (almost 68%), whereas Poland is a relatively closed economy (less than 40%). This is likely because the Czech Republic, Slovakia, and Hungary are dependent on foreign trade, but in the case of Poland, this country is relatively closed, but because of low value of GDP, other parts of the GDP formula are more important than foreign trade.

### 3.4. Foreign Trade Development—Trade with Services

The analysis of foreign trade with services shows completely different results. All V4 countries have been in surplus, with only a few exceptions in the case of Slovakia. However, the development is quite irregular in all countries, with several increases and decreases. Surpluses in the Czech Republic and in Slovakia decreased, while surpluses in Poland and in Hungary increased.

Interestingly, in 2009, where GDP in all analysed countries dropped, the net balance with services dropped as well, where trade balances in terms of goods increased in all analysed countries. The biggest net balance per capita was Hungary, the Czech Republic is in second position, Poland is third, and Slovakia is in last position. On the other hand, Slovakia has the highest growth rate (more than 300%), but it is probably also because of low level of net balance. Interestingly, Hungary, even if it has the highest net balance per capita, also has a growth rate over 100%.

The highest export of services/GDP share is Hungary, where this ratio is almost 19%. The Czech Republic has this share currently around 12.4%, and the other countries less than 10%. However, the development in all analysed countries has been more or less irregular. Figure 3 describes again the relation between the values of net balance per capita in 2016 and the growth rate between 2011 and 2016.

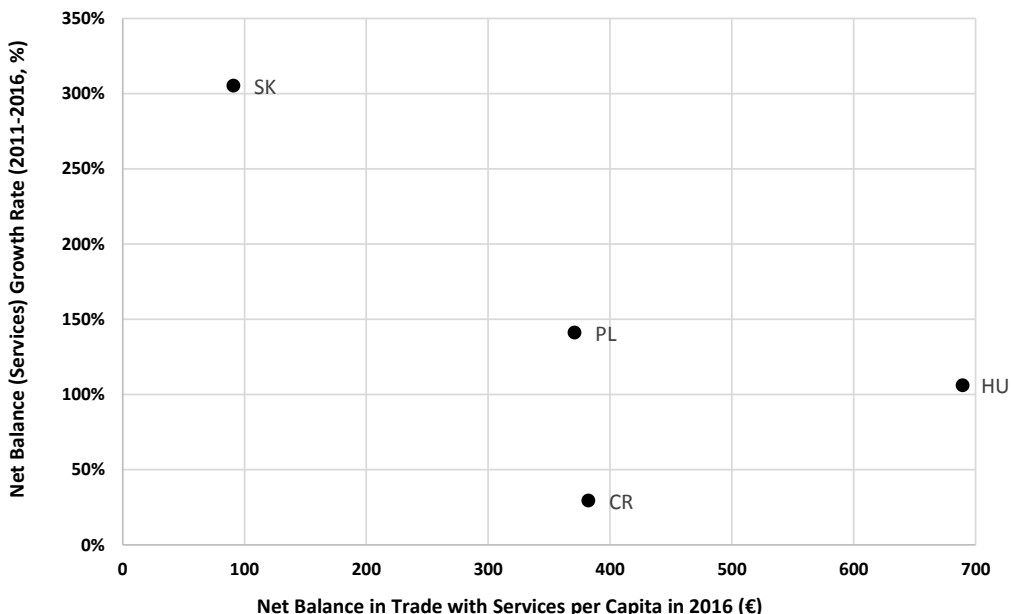

**Figure 3.** Relation between net balance in trade with services per capita (2016) and growth rate (2011–2016).

### 3.5. Nonmetric Scaling of Visegrad Four Countries

As was mentioned in the methodology, the method of point rating has been used for this analysis. Three different macroeconomic indicators have been used for the rating, namely GDP per capita in Euro (A); balance of goods per capita in Euro (B); and balance of services per capita in Euro (C). The analysed period for this analysis is 2000–2016, because it is possible to find enough data for such analysis (unfortunately, for GII and SII development analysis it is not). However, the years 2011 and 2016 are compared in the analysis, where points in every country for every analysed indicator have been calculated and summed up.

Figure 4 evaluates V4 countries from the perspective of GDP per capita and foreign trade per capita, separately foreign trade with goods and with services. In the analysed period 2011–2016, the Czech Republic and Hungary have had more points than Slovakia or Poland. On the other hand, Slovakia and Poland have had the higher growth rate of these points (probably because of low values of points). Moreover, the analysis shows that all countries except Slovakia continually have grown in

terms of points during the analysed period (2011–2016), where Slovakia had been losing its position since 2013. This is because the foreign trade in Slovakia has not had as a significant impact on GDP as in the other V4 member states, especially in the Czech Republic and in Hungary. Foreign trade has the positive effect on growing GDP per capita in all the analysed countries, but it is possible to identify significant differences among V4 members.

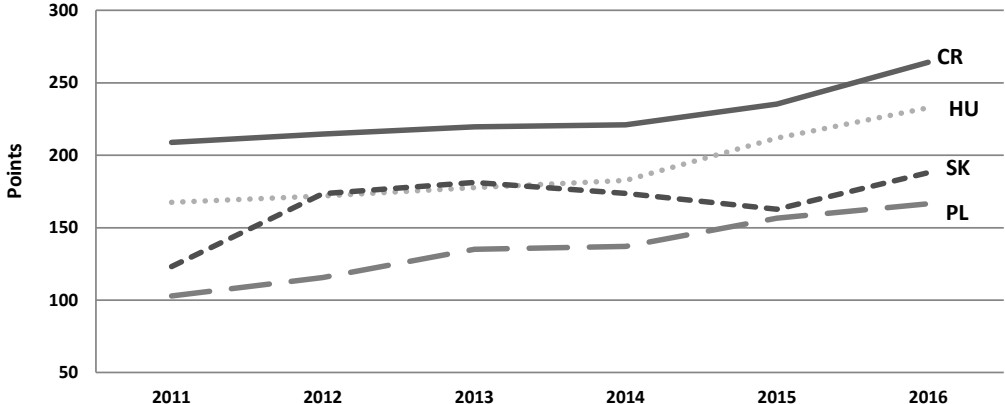

**Figure 4.** Development of points in 2011–2016 (years, points); Point are calculated together for GDP per capita in €, Balance of goods per capita in €, and Balance of services per capita in €.

### 3.6. Innovativeness of the Visegrad Four Countries—GII and SII Indicator

GII data show that in 2016, the innovation of the V4 countries was low. The value of the innovation index of all the Visegrad Group countries was below the EU average. The innovation rate in these countries did not exceed 50 points out of 100 possible. Poland and Slovakia have adopted the lowest values of innovation rate among EU countries and found themselves in the weakest group of so-called "catching-up countries". The higher value of GII, although below the average, was recorded by the Czech Republic and Hungary. These countries were in the so-called "moderate innovators" group (Figure 5, based on Dutta and INSEAD 2011; INSEAD 2011; Dutta et al. 2016; Cornell University et al. 2016; World Economic Forum 2017).

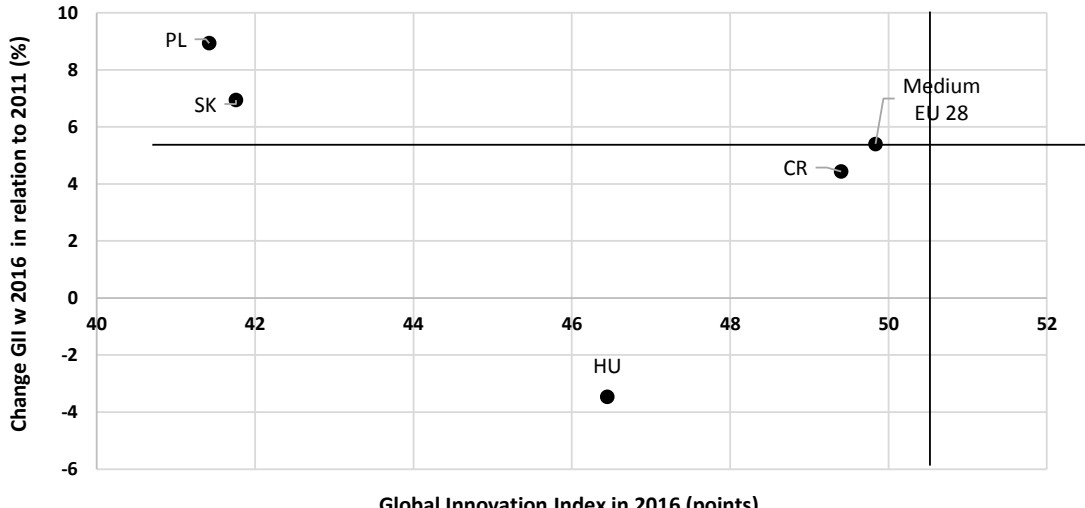

**Figure 5.** Global Innovation Index (GII) in 2016 and change over the period of 5 years (2011–2016) in Visegrad Four (V4) countries (points, %).

The European Institute of Management provides data on innovativeness of economies, taking into account the division into indicators of innovative position and ability to innovate

(Dutta and INSEAD 2011; Dutta et al. 2016). The analysis of the Institute's data on the V4 countries showed both differences in the level of innovation in the countries discussed, as well as significant changes in some of them between 2011 and 2016 (Table 2, based on Dutta and INSEAD 2011; INSEAD 2011; Dutta et al. 2016; Cornell University et al. 2016; World Economic Forum 2017).

**Table 2.** Innovation rating, innovation output and innovation input in V4 countries in 2016 compared to 2011.

| Country | Innovation Output Sub-Index | | | Innovation Input Sub-Index | | |
|---|---|---|---|---|---|---|
| | 2011 | 2016 | Change (%) | 2011 | 2016 | Change (%) |
| Czech Republic (CR) | 41.5 | 44.5 | 7.3 | 53.1 | 54.3 | 2.3 |
| Hungary (HU) | 45.2 | 40.5 | −10.4 | 51.0 | 48.9 | −4.1 |
| Slovakia (SK) | 29.8 | 35.4 | 18.7 | 48.3 | 48.0 | −0.6 |
| Poland (PL) | 29.7 | 31.7 | 6.6 | 46.3 | 48.7 | 5.2 |

The V4 countries are not among the innovation leaders in the world. The Czech Republic, among the countries discussed, occupied the highest position in the Global Innovation Index and improved (over five years) both the value of the sub-index of innovative contributions (by 2.3%) as well as the sub-index of innovative results (by 7.3%). Lower indices were visible in the case of Hungary: the sub-index of innovative contributions (by 4.1%) and the sub-index of innovative results (by 10.4%). In the analysed period, Poland's ratios increased significantly, although they still had the lowest values among the V4 countries. In 2016, the sub-index of innovations for Poland amounted to 31.7 points (increase by 6.6% in five years). In the case of the sub-index of the innovative contribution, Poland overtook Hungary by 0.7 points.

Also in the reports European Innovation Scoreboard (EIS) proposed by the European Commission, the value of SII for all countries of the Visegrad Group in both 2011 and 2016 was below the EU average (Figure 6, based on Dutta and INSEAD 2011; Dutta et al. 2016; European Union 2016). In the case of SII reports, all V4 countries were included in the so-called "moderate innovators" group. In 2016, the value of the SII index for the Czech Republic was 0.42 and was the highest among the V4 countries. The last position was taken by Poland and Hungary, for whom the SII index value was 0.27. The value of SII index in 2016 in relation to 2011 increased in the case of Slovakia (by 4.9%) and Poland (by 2.7%), the decrease was recorded by the Czech Republic (by 5.2%) and Hungary (by 2.2%).

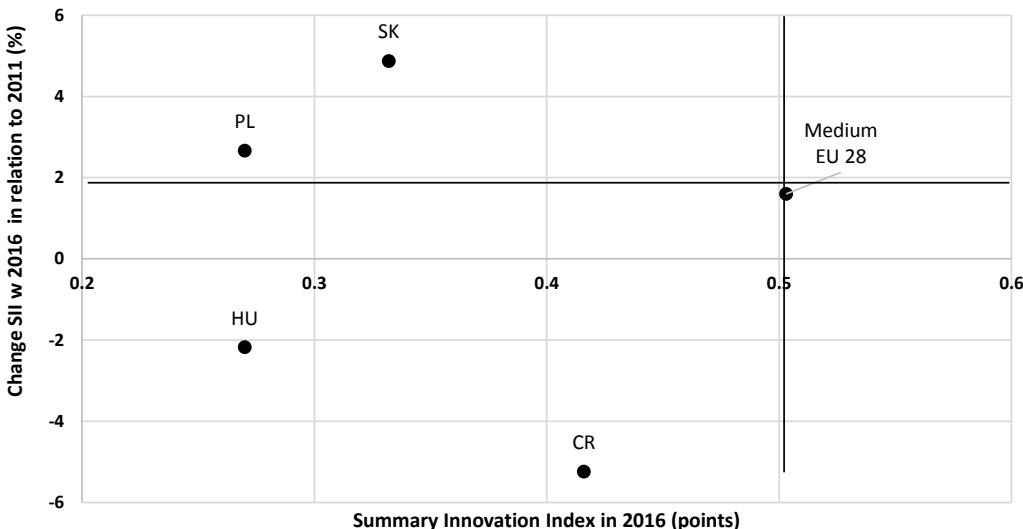

**Figure 6.** Summary Innovation Index of the EU 28 and V4 countries in 2016 year and change in change in relation to 2011 (points, %).

Analysis of the Visegrad Group countries data according to EIS reports from 2011 and 2016, divided into 4 groups of indicators showed (similarly as in the case of GII) differences both in the level of innovativeness of the countries studied and significant changes that took place during the analysed period (Table 3, based on European Union 2016, European Innovation Scoreboard (EIS) (2017)).

**Table 3.** Innovativeness ranking by four thematic groups EU and V4 in 2016 as compared to 2011 (points, %).

| Country | Framework Conditions | | | Investments | | | Innovation Activities | | | Impacts | | |
|---|---|---|---|---|---|---|---|---|---|---|---|---|
| | 2011 | 2016 | Change (%) | 2011 | 2016 | Change (%) | 2011 | 2016 | Change (%) | 2011 | 2016 | Change (%) |
| EU 28 | 0.5 | 0.48 | −4.0 | 0.42 | 0.48 | 14.3 | 0.49 | 0.46 | −6.1 | 0.52 | 0.48 | −7.7 |
| Czech Republic (CR) | 0.29 | 0.38 | 31.0 | 0.55 | 0.44 | −20.0 | 0.41 | 0.34 | −17.1 | 0.64 | 0.56 | −12.5 |
| Slovakia (SK) | 0.3 | 0.3 | 0.0 | 0.25 | 0.36 | 44.0 | 0.23 | 0.22 | −4.3 | 0.62 | 0.64 | 3.2 |
| Hungary (HU) | 0.26 | 0.3 | 15.4 | 0.27 | 0.29 | 7.4 | 0.27 | 0.2 | −25.9 | 0.71 | 0.66 | −7.0 |
| Poland (PL) | 0.19 | 0.27 | 42.1 | 0.29 | 0.3 | 3.4 | 0.21 | 0.18 | −14.3 | 0.47 | 0.41 | −12.8 |

In 2016, the Czech Republic, in comparison with other V4 countries, obtained the highest values in three out of four thematic groups, i.e., framework conditions, investments and innovative activities. The lowest values of the indicators in three of the four groups were obtained by Poland, while in the case of the fourth indicator—"Investments" this country overtook Hungary by only one point. The remaining countries of the Visegrad Group recorded an increase in the value of the index in the investment area—the highest was achieved by Slovakia (by 44%). In the group framework conditions, the index increased in Poland (by 42.1%), in the Czech Republic (by 31%) and in Hungary (15.4%). Slovakia, as the only one, maintained its value from 2011. In all V4 countries, the index in the group of innovative activities decreased. The largest decrease was recorded, in addition to the mentioned Czech Republic, also in Hungary. In the last group of "Impacts", three from V4 countries, i.e., Poland, the Czech Republic and Hungary, recorded a decrease. The highest value of the indicator in 2016 (despite a decrease by 7%) was obtained by Hungary. In the case of Poland and the Czech Republic, the index decreased by 12.8% and 12.5% respectively. Slovakia was the only one that recorded growth in the Impacts ratio by 3.2%.

*3.7. Competitiveness of the Visegrad Four Countries—GCI Indicator*

According to the Global Competitiveness Report for 2015–2016, the countries of the Visegrad Group were qualified (as in the case of GII index) into two different groups: Czech Republic and Slovakia to the so-called group advanced economies and Poland and Hungary to the "emerging and developing Europe" group (Schwab 2015).

In 2015–2016, the highest value of the GCI index among the V4 countries was recorded by the Czech Republic, just as in the case of previous indicators. This country has increased the value of GCI index, compared to 2011, by 2.2% (Figure 7, based on Schwab 2010, 2015). Hungary, in the same period, recorded a decrease of its value by 2.3%. On the other hand, the other countries, i.e., Poland and Slovakia, have maintained GCI value at the same level.

The analysis of the competitiveness level of the economies of the Visegrad countries divided into three groups of indicators also showed equations between the surveyed countries (Table 4, based on Schwab 2010, 2015).

In 2010–2011 and 2015–2016 the Czech Republic, in comparison with other V4 countries, obtained the highest values of indices in all sub-indices. In the years 2015–2016, the values of individual sub-indices for the Czech Republic were as follows: basic requirements—5.27 points, efficiency enhancers—4.78 points and innovations and sophistication factors—4.14 points. The lowest index values in all three sub-indices in the years 2015–2016 were obtained by Hungary.

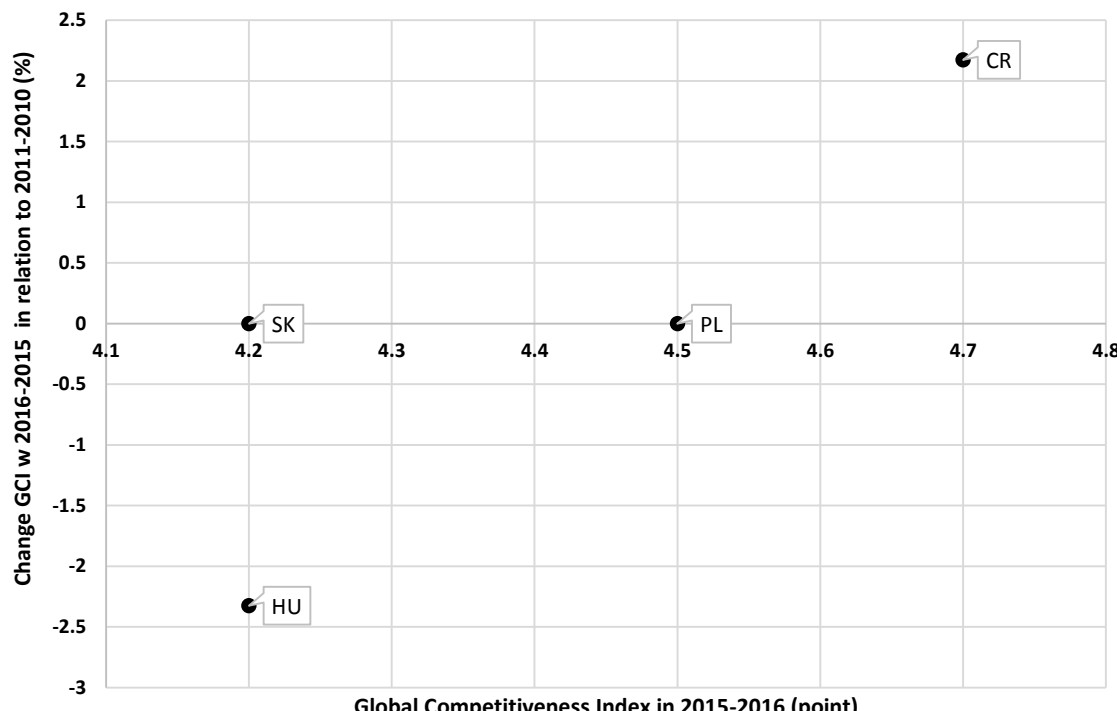

**Figure 7.** Global Competitiveness Index in V4 countries in 2010–2011 and 2015–2016 (points, %).

**Table 4.** The competitiveness index in V4 country by three sub-indexes in 2015–2016 as compared to 2010–2011 (points, %).

| Country | Basic Requirements | | | Efficiency Enhancers | | | Innovation and Sophistication Factors | | |
|---|---|---|---|---|---|---|---|---|---|
| | 2010–2011 | 2015–2016 | Change (%) | 2010–2011 | 2015–2016 | Change (%) | 2010–2011 | 2015–2016 | Change (%) |
| Czech Republic (CR) | 4.91 | 5.27 | 7.33 | 4.66 | 4.78 | 2.58 | 4.19 | 4.14 | −1.19 |
| Slovakia (SK) | 4.77 | 4.73 | −0.84 | 4.43 | 4.34 | −2.03 | 3.54 | 3.68 | 3.95 |
| Hungary (HU) | 4.65 | 4.67 | 0.43 | 4.38 | 4.31 | −1.60 | 3.71 | 3.57 | −3.77 |
| Poland (PL) | 4.69 | 4.91 | 4.69 | 4.62 | 4.64 | 0.43 | 3.76 | 3.70 | −1.60 |

The analysis of sub-indices in V4 showed that in the years 2015–2016 the Czech Republic increased the value of followed sub-indexes: "basic requirements" (by 7.33%) and "efficiency enhancers" (by 2.6%). On the other hand, the sub-index "factors of innovation and sophistication" dropped (by 1.2%). The increase in value of "basic requirements" sub-index was also recorded in Poland (by 4.7%) and Hungary (by 4.3%). In the case of Slovakia, its value decreased slightly (by 0.82%). In the case of the "efficiency enhancers" sub-index, the growth in its value was recorded, apart from the Czech Republic, also in Poland. However, the increase in the value of this indicator in the case of Poland was insignificant (only 0.4%). The other two countries, i.e., Slovakia and Hungary, recorded a decrease in the value of this sub-index. In the last sub-index, which is "factors of innovation and sophistication" only Slovakia in the analysed period obtained an increase (by 4%). This allowed Slovakia to be promoted in the ranking to the third position. Other countries recorded a decrease in this indicator; the highest in the case of Hungary amounted to 3.8%.

*3.8. The Relation between GDP and Innovation Indexes (GII, SII, GCI)*

The mutual relation between GDP development and GII, SII, and GCI development is analysed in this section, where all these indicators have been already analysed separately in previous sections. Figure 8 describes the position of every analysed country in 2011 and 2016, where it is possible to compare not only the development, but also the size of the country (thanks to the size of every bubble).

Figure 8 describes the mutual relation between GDP and each index, where the size of the bubble (for better visualization) presents the size of the country measured by the number of inhabitants.

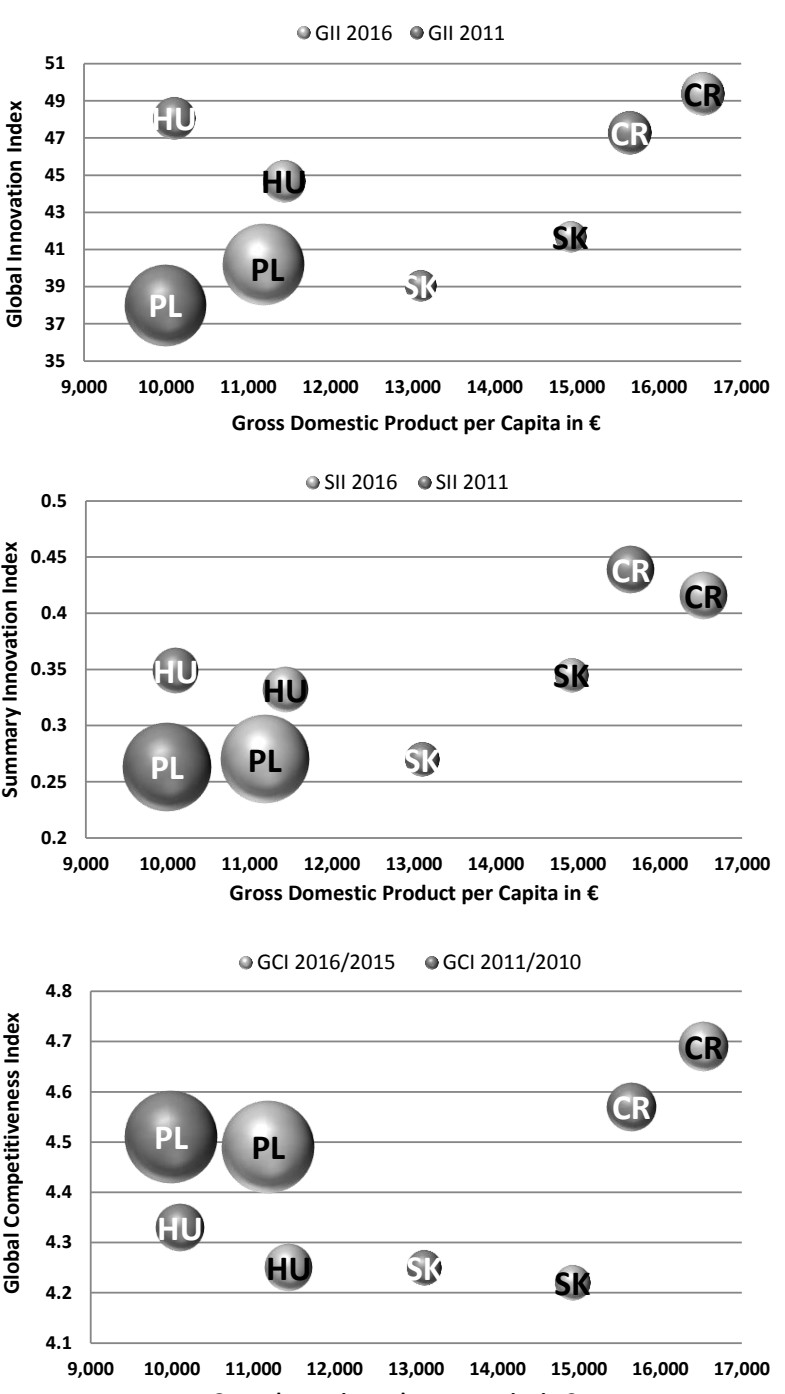

**Figure 8.** Relation between GDP per capita and GII (**A**); Summary Innovation Index (SII) (**B**); and Global Competitiveness Index (GCI) (**C**).

It is possible to see that even if every V4 member has recorded significant increase in terms of GDP per capita development between 2011 and 2016, this increase was not followed by the increase of each innovation index, namely by Global Innovation Index, Summary Innovation Index, and Global Competitiveness Index.

The biggest increase of GDP per capita has been in Slovakia (13.91%), where GII and SII have increase as well, but GCI has decreased in this country (2016 compared to 2011). Hungary has had the increase of GDP per capita for 13.24%, but all indexes have dropped. Poland has had the increase of GDP per capita for 11.94%, but the positive development was only in case of GII, where other indicators either have remained almost same or have decreased. The Czech Republic has a unique position among V4. The absolute value of GDP per capita is the highest in this country, where this indicator has shown change only for 5.67% (the lowest in V4), GII and GCI have increased, but SII has decreased.

Based on these facts is not possible to find any relation between the growth of every V4 country (measured by GDP per capita), and the development of Global Innovation Index, Summary Innovation Index, and Global Competitiveness Index. These indexes include a lot of partial evaluated criterion, which did not have the impact on economic growth of each country during evaluated period.

## 4. Conclusions

The article analyses selected aspects of the Visegrad Four Countries, namely GDP development, foreign trade, innovation, and competitiveness indicators of the V4 economies. The authors are aware of different aspects of global development, as well as different influences affecting GDP development; however, the aim of this article was to discover the relationship between GDP development, foreign trade development, and innovative potential. On the basis of the comparison of V4 countries' indices, it can be seen that they differ both in terms of innovation and competitiveness.

The analysis of the GDP development has shown the strong position of the Czech Republic among V4 countries, but this strong position is getting weaker every year because of the slow growth rate of this country after the global economic crisis in 2008. Slovakia, especially, is getting closer to the Czech Republic.

Foreign trade is very important for almost every country around the world. Not only does it solve the proportionality problem, but it can also either increase or worsen the level of GDP, because it is part of the GDP formula in open economies. All the member states of the V4, except Poland, can be considered as open economies; however, the analysis of net balance with trade with goods is showing relatively different results than the GDP analysis. The Czech Republic is again the strongest economy with the highest net balance per capita, but it has also the second highest growth rate. Significantly highest growth rate is Slovakia, but this country has the third lowest level of net balance per capita.

Trade with services also shows some surprising results. The highest value of net balance per capita is Hungary, where the growth rate is also over 100%. The highest growth rate is Slovakia (more than 300%), but it is probably only because of the lowest value of net balance per capita in this country.

The GII and SII innovation reports presented in the article either put all the Visegrad countries in one group (so-called moderate innovators—SII report) or divided them into two different groups: the Czech Republic and Hungary to moderate innovators, Poland and Slovakia to so-called "catching-up countries" (GII report). The higher value of GII was recorded in the Czech Republic and Hungary. The value of GII shows that all the V4 countries in 2016 had values below the average for the European Union. On the other hand, the SII index values obtained by these V4 countries were, in the analysed years, in the range of 50–90% of the EU average.

The results from the Global Index Innovation and from the European Innovation Scoreboard report from 2011–2016 show a growing gap in the level of innovation between the V4 countries. According to SII reports, the Czech Republic is beginning to move away in terms of innovation from other V4 countries. The Czech Republic is catching up with the countries from the so-called supporters of innovation group, while Poland is moving towards the so-called overtaking countries. Poland occupies one of the last places in innovation rankings (this is evidenced by both GII and SII values) among the V4 countries in general as well as in individual thematic groups.

Also, the analysis of the competitiveness of the V4 economies showed a significant difference between the countries surveyed. The Czech Republic, which is increasing its GCI index year by year,

is the most competitive than the other V4 countries. In 2016, Hungary was the weakest in terms of competitiveness. The value of competitiveness indices for this country in the period covered by the study has deteriorated even further, whereas Slovakia is trying to maintain the unchanged index value.

Analysis of innovation and competitiveness indicators shows that V4 countries are beginning to differ more and more from each other. Despite the adoption of a different methodology in the discussed indices, one notices in all cases, the growing advantage of the Czech Republic (regardless of the type of report). Certainly, the increase in innovation and competitiveness of individual economies (including the Czech Republic) would not be possible without adequate financial resources. Only countries that are able to obtain and then properly use their financial resources (e.g., EU subsidies) are able to improve their market position. From the obtained results it can be concluded that the strategy adopted by the Czech Republic was correct. It is a pity that the remaining countries of the Visegrad Group failed to increase their innovation and competitiveness indices to get closer to the countries of the old European Union.

However, mutual comparison between GDP development and development of analysed indexes (GII, SII, and GCI) has shown no relation. These indexes include many partial evaluated criteria, which do not have the impact on economic growth of each country during evaluated period.

**Author Contributions:** Data calculation, A.K., J.K., E.H., P.P.; Formal analysis, A.K., J.K., E.H. and P.P.; Methodology, A.K., J.K., E.H. and P.P.; Writing—original draft, A.K., J.K., E.H., P.P.; Writing—review and editing, A.K., J.K., E.H. and P.P.

**Conflicts of Interest:** The authors declare no conflicts of interest.

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
