# Peer review of "The Selected Topics for Comparison in Visegrad Four Countries"

_economies, doi:10.3390/economies6030050_

Round 1

Reviewer 1 Report

Even if the authors sought to address some of my earlier suggestions, my biggest concern pertains to the choice of literature: seemingly the authors cite themselves extensively, which does not validate the argument; quite the contrary. 

Moreover, also as for the soundness of the argument, I am not convinced that an analysis of selected topics, without considering others which can indeed dramatically change the outcomes of the study, can be an explicit focus of a paper. 

Even if I believe the topic is very salient, I lack depth in the quality of the argumentation. For reference, I would suggest that the authors consider the following book (not, mine): Farkas, B. (2016) Models of Capitalism, publication https://www.palgrave.com/gp/book/9781137600561

The author offers a very good analysis that covers also the V4 countries. Maybe the authors of this paper would like to consider it. 

Author Response

We would like to thank to the reviewer about his suggestions. Firstly, the methodology part of the article has been enlarged by following text:

"A lot of authors of scientific publications compare different criterion in different countries with the aim to discover mutual consistency, to identify differences, and to infer conclusions about possible future tendencies in global economy development. For example, using the Central and Eastern European model of capitalism, paper by Farkas (2017) compares the market economies of the Western Balkan countries to the post-socialist European Union member states. It analyses the main institutional areas of a socio-economic system, such as product markets, innovation system, financial system, labour market and industrial relations, social protection and the educational system. The paper by Fagerberg, Srholec (2007) outlines a synthetic framework, based on Schumpeterian logic, for analysing the question why do some countries perform much better than other countries. Four different aspects of competitiveness were identified: technology, capacity, demand, and price. The contribution of the paper was particularly to highlight the three first aspects, which often tend to be ignored due to measurement problems. The empirical analysis, based on a sample of 90 countries on different levels of development during 1980–2002, demonstrated the relevance of technology, capacity, and demand competitiveness for growth and development. Price competitiveness seemed generally to be of lesser importance. Since the collapse of state socialism in the late 1980s, the Czech Republic, Hungary, Poland, and the Slovak Republic have introduced a rather successful model of capitalism when compared with other post-socialist states. The article by Nölke and Vliegenthart (2009) identifies the key elements of the Dependent Market Economy (DME) model and discusses their interplay. DMEs have comparative advantages in the assembly and production of relatively complex and durable consumer goods. These comparative advantages are based on institutional complementarities between skilled, but cheap, labour; the transfer of technological innovations within transnational enterprises; and the provision of capital via foreign direct investment. Increased openness has also resulted in a growing inflow of FDI, both as a result of large-scale privatization programs and green-field investment by Cernat (2006). Authors Blanke, Hoffmann (2008) outline the five pillars of an upcoming European social model, although they deny the existence of a closed system of social and economic regulatory mechanisms designed to correct the operation of markets. Yet, what exists in all western European member states is a comparatively high level of welfare state protection, and, at the European level, a certain number of elements for the development of a specific welfare state framework has been established.

The authors of this article evaluate macroeconomic indicators as categories related to the economic power of each analysed country. One of the most important indicators is foreign trade with goods and services and its influence on global economic development of V4 countries. Inseparable part of comparative analysis is innovative potential of analysed countries, where this potential is evaluated by the change of Global innovative index (GII), and by the change of Global competitiveness index (GCI)."

Secondly, the literature has been enlarged by following articles:

   1.    Blanke, T., Hoffmann, J. (2008). Towards a European social model preconditions, difficulties and prospects of a European social policy. International Journal of Public Policy, 3 (1-2), 20-38. DOI: 10.1504/IJPP.2008.017124

   2.    Cernat, L. (2006). Institutions and Economic Growth in Central and Eastern Europe: A Quantitative Analysis in Europeanization, Varieties of Capitalism and Economic Performance in Central and Eastern Europe, Basingstoke: Palgrave Macmillan. DOI:10.1057/9780230501683_3

   3.    Fagerberg, J., Srholec, M., Knell, M. (2007). The competitiveness of nations: Why some countries prosper while others fall behind? World Development, vol. 35 (10). 1595-1620. DOI: 10.1016/j.worlddev.2007.01.004

   4.    Farkas, B. (2017). Market Economies of the Western Balkans Compared to the Central and Eastern European Model of Capitalism. Croatian Economic Survey, vol. 19 (1). 5-36. DOI: 10.15179/ces.19.1.1

   5.    Nolke, A., Vliegenthart, A. (2009). Enlarging the Varieties of Capitalism. The Emergence of Dependent Market Economies in East Central Europe. World Politics, vol. 61(4), 670–702. DOI: 10.1017/S0043887109990098.

Reviewer 2 Report

I recognize to the authors their maximum effort to improve the paper.

Author Response

Thank you very much, some improvements in Methodology and in the Literature have been done.

Round 2

Reviewer 1 Report

Thank you for addressing my suggestions.